# OpenReview forum: "IDEATOR: Jailbreaking VLMs Using VLMs"
_ICLR.cc/2025/Conference — ICLR 2025 Conference Withdrawn Submission_

### Official Review · Reviewer_TB9E · 2024-10-27

**Soundness:** 2
**Presentation:** 3
**Contribution:** 1
**Rating:** 3
**Confidence:** 4

**Summary:**

This paper designs a new jailbreak attack targeting Vision-Language Models. This attack method utilizes a VLM to simultaneously generate text and image prompts, and then generate images with an SD model. The text prompt and image are fed into the victim VLM, and based on its response, optimization is performed to find prompts that can jailbreak. The method is instruction-based and thus training-free, making it easy to implement.

**Strengths:**

- The paper is well-written and easy to follow.
- The method is training-free and easy to implement.
- The attack is quite effective compared to baselines.

**Weaknesses:**

- Although the method is effective, it is merely a prompt engineering work and does not bring any technical or theoretical innovation. Therefore, the contribution of this work is insufficient.
- It did not test closed-source commercial models like GPT-4o, making the results less convincing.
- While the work uses a breadth-depth exploration strategy, it is not a new technique. Moreover, the authors did not explain or demonstrate the diversity changes brought by this strategy. The current results only show that increasing depth and breadth can improve the ASR, which does not necessarily indicate diversity.
- Although the authors presented some findings about the IDEAtor in the Empirical Understanding section, they did not explain why the attacker VLM exhibits these behaviors. This makes it difficult to transfer this method to other models.
- The authors did not carefully explain what changes in the "analysis" enable the attacker VLM to continuously optimize malicious prompts.
- Many detailed experimental results were not presented, making the contribution of this paper limited.

**Questions:**

- How are the relevant examples in the attacker VLM defined? Do they need to be designed differently for various SD models, attacker VLMs, and victim VLMs?
- If the attacker VLM can analyze text and images, does it need to have capabilities approximately equal to the victim VLM? Why or why not?
- As mentioned, the attacker VLM may refuse to output malicious text or image prompts. Besides using MiniGPT-4, how did the authors solve this problem? Because larger victim VLMs require stronger attacker VLMs, relying solely on MiniGPT-4 as the attacker is obviously insufficient.
- Do the text prompts or image prompts contain harmful tokens? Do the images generated by SD contain harmful content? If so, will they be filtered out by the victim VLM or SD’s safety measures?
- Are the text prompts and generated images semantically related? How is this relevance controlled?
- How do you address the randomness of Stable Diffusion (SD)? Can all images generated from a success image prompt successfully perform the attack?

**Details Of Ethics Concerns:**

Since the paper focuses on the jailbreak to VLM, discussion about ethical concerns should appear in the draft.

---

### Official Review · Reviewer_ZQFN · 2024-10-31

**Soundness:** 3
**Presentation:** 2
**Contribution:** 3
**Rating:** 5
**Confidence:** 3

**Summary:**

This paper introduces IDEATOR, a novel black-box jailbreak method for Vision-Language Models (VLMs) that addresses limitations in current approaches to VLM safety evaluation. Unlike existing methods that rely on adversarial or query-relevant images from harmful text datasets, IDEATOR innovatively transforms a VLM into a jailbreak agent through system prompts and conversation templates, working in conjunction with a diffusion model to autonomously generate malicious image-text pairs. The method's effectiveness is demonstrated through extensive experiments, achieving remarkable success rates (94% on MiniGPT-4, 82% on LLVA, and 88% on InstructBLIP), revealing significant vulnerabilities in current VLMs and highlighting the need for more robust safety mechanisms. This work not only presents a powerful tool for red teaming but also emphasizes the potential risks of using VLMs themselves as adversarial agents.

**Strengths:**

1. The paper presents a novel and innovative approach by leveraging VLMs as adversarial agents.
2. The methodology is well-structured and clearly articulated.
3. The experimental results demonstrate impressive effectiveness and transferability across different VLM architectures.
4. The work provides valuable insights into VLM vulnerabilities and security implications.

**Weaknesses:**

1. Technical Documentation:
   - Typographical error: "llva" should be "llava" (line 25)

2. Evaluation Methodology Concerns:
   - The reliance on manual review of harmful outputs requires additional validation
   - Comparison with automated evaluation methods (keyword matching and LLM-based evaluators) would strengthen the methodology's credibility
   - The results from these alternative evaluation approaches should be included for completeness

3. Experimental Design Limitations:
   - The use of identical models (Vicuna-13B version of MiniGPT-4) for both attacker and victim roles raises potential bias concerns
   - Direct transfer attack results between different architectures (e.g., MiniGPT-4 as attacker → LLaVA as victim) need to be explicitly demonstrated

**Questions:**

1. Could the authors provide comparative results between manual reviews and automated evaluation methods to validate the assessment methodology?
2. What are the performance metrics when using different model architectures for attacker and victim roles, particularly in the MiniGPT-4 → LLaVA scenario?
3. How might the use of identical models for attack and defense impact the generalizability of the results? I would consider decrease the ratings if no enough experiments to solve my questions.

---

### Official Review · Reviewer_c4V7 · 2024-11-02

**Soundness:** 2
**Presentation:** 2
**Contribution:** 1
**Rating:** 3
**Confidence:** 5

**Summary:**

This paper introduces IDEATOR, an innovative black-box jailbreak framework designed specifically for large Vision-Language Models (VLMs). IDEATOR overcomes the limitations of existing jailbreak techniques by autonomously generating varied image-text pairs, transforming the VLM into an active jailbreak agent when paired with a diffusion model. The approach establishes a new paradigm in multimodal jailbreak prompts by simulating adversarial user behavior, iteratively evolving jailbreak strategies to expose the multimodal vulnerabilities of VLMs. Experimental results validate IDEATOR’s high effectiveness, with a notable 94% success rate in jailbreaking MiniGPT-4 and significant transferability, achieving high success rates across other models such as LLVA and InstructBLIP.

**Strengths:**

The originality of this work presents some interesting aspects, particularly in its attempt to use a VLM to generate jailbreak prompts for attacking other VLMs. While the concept of leveraging both a visual language model and a diffusion model is notable, it remains to be seen how effectively this approach can uncover the vulnerabilities of VLMs compared to existing methods. The extension of red-teaming from LLMs to multimodal systems is a step forward, yet it raises questions about its practical implications and effectiveness.

Regarding the quality of the experimental setup, it is somewhat comprehensive but could benefit from further refinement. The inclusion of datasets like Advbench and VAJM, along with metrics such as the Attack Success Rate, adds a layer of evaluation; however, the overall rigor is limited by the lack of clarity on certain implementation details. While the specified models and exploration parameters provide some reproducibility, more thorough ablation studies would be necessary to fully understand the impact of various factors on attack performance. Overall, the study has potential, but significant improvements are needed for it to be considered robust.

**Weaknesses:**

1.The experiments conducted primarily focus on a narrow set of VLMs, specifically MiniGPT-4, LLaVA, and InstructBLIP. This limited selection may not accurately reflect the broader landscape of VLMs, potentially skewing the conclusions about the generalizability and effectiveness of IDEATOR. Including a wider variety of victim models with different architectures would provide a more comprehensive understanding of how the proposed method performs across different contexts.

2.The theoretical justification for utilizing a VLM as a red team model for generating jailbreak prompts is weak and lacks a solid foundation. While empirical results are presented, there is no comprehensive theoretical framework explaining how these models can generate effective prompts. Additionally, the paper does not address potential biases in the attacker and victim VLMs, which could significantly influence the results.

3.The overall innovation of this work is insufficient, particularly in its similarity to existing research, such as the method outlined in Reference 1. The authors should provide a detailed explanation of how their approach differentiates from that prior work. Furthermore, the lack of plans to release the code and data limits reproducibility, and the paper does not sufficiently discuss the real-world applicability of its findings, which is crucial for understanding the broader impact of these vulnerabilities.

Ref1:https://arxiv.org/abs/2407.15050

**Questions:**

1.The experiments in this paper focus primarily on a narrow selection of VLMs, specifically MiniGPT-4, LLaVA, and InstructBLIP. This limited scope raises concerns about the generalizability of the findings. There are numerous other VLMs with varying architectures and characteristics that could yield different responses to the proposed jailbreak attacks. The authors should clarify why these particular models were chosen and discuss how representative they are of the wider VLM population. Including a broader set of victim models would enhance the validity of the results and provide a clearer picture of IDEATOR's performance across different types.

2.While the black-box nature of IDEATOR is a significant aspect of the study, the paper does not adequately explore the potential benefits of white-box attacks. Understanding how IDEATOR performs in a white-box context could provide valuable insights into the strengths and weaknesses of the approach. The authors should consider discussing the trade-offs between black-box and white-box methods in terms of effectiveness and practicality. Additionally, the lack of exploration into hybrid attack strategies limits the understanding of VLM vulnerabilities. A discussion on how a hybrid approach could be designed and evaluated would add depth to the analysis.

3.The conjecture that a VLM can serve as an effective red team model for generating jailbreak prompts lacks a solid theoretical underpinning. Although the empirical results suggest effectiveness, there is insufficient theoretical analysis to explain why a VLM might generate effective jailbreak prompts. The authors should explore relevant concepts from cognitive science, linguistics, or machine learning theory to create a more robust theoretical framework. This would not only strengthen the claims made in the paper but also provide a deeper understanding of the mechanisms at play.

4.The paper neglects to account for potential biases in both the attacker and victim VLMs. These biases could significantly influence the generation and effectiveness of the jailbreak prompts in ways that are not fully understood. The authors should address how biases in the attacker VLM might lead to an incomplete exploration of jailbreak strategies. Furthermore, exploring how the biases in the victim VLM affect its vulnerability to different types of attacks is crucial. A discussion on the role of model biases in the jailbreak process, along with strategies for mitigation or exploitation, would add valuable insight.

Overall, the study would benefit from a more rigorous methodological approach. The limited diversity in victim models, the lack of exploration of different attack strategies, and the need for a stronger theoretical foundation all point to areas that require further attention. Enhancing the study's methodological rigor would not only improve the credibility of the findings but also contribute to a more comprehensive understanding of the implications of IDEATOR in the context of VLM vulnerabilities.

---

### Official Review · Reviewer_TP2C · 2024-11-04

**Soundness:** 3
**Presentation:** 4
**Contribution:** 3
**Rating:** 5
**Confidence:** 5

**Summary:**

This paper presents a jailbreaking attack method called IDEATOR, which generates more diverse jailbreaking images. Specifically, the method employs a VLM to generate jailbreaking texts and uses state-of-the-art diffusion models to create jailbreaking images. Experiments demonstrate the effectiveness and transferability of this attack.

**Strengths:**

1. The authors present a novel VLM jailbreak attack method capable of creating more diverse jailbreak images.
2. The paper is well-written and easy to read.

**Weaknesses:**

1. Exaggeration in claims. The abstract claims to have discovered previously unexplored vulnerabilities in VLMs; however, this is still a VLM jailbreak attack, and the effectiveness of multi-round iterations in increasing attack success rates has already been validated in other LLM jailbreak scenarios. The authors should avoid overstating this aspect.
2. Lack of analysis on query counts. The method improves attack success rates through multiple rounds of queries, which is understandable. However, how does this attack's query and time costs compare to other methods? A more detailed analysis and discussion are needed. In a black-box setting, excessive queries can make the attack more detectable.
3. Limited discussion on model selection. Different VLM models have varying generative capabilities, and choosing only MiniGPT-4 to demonstrate the method's effectiveness and generalizability does not prove its broad applicability. The authors should discuss a wider range of attackers and victims to validate the effectiveness of the method.
4. Lack of discussion on defenses and countermeasures. Various jailbreak defense methods for LLMs or VLMs have been proposed, and the authors should assess the performance of their method under such defenses. Additionally, as an attack paper, the authors do not provide any insights on potential defenses.

**Questions:**

Please refer to the weakness.

---

### Note · Authors · 2024-11-12

I have read and agree with the venue's withdrawal policy on behalf of myself and my co-authors.